

# The mediation role of sleep on the relationship between drinks behavior and female androgenetic alopecia

Shiqi Liu[1,2,*], Hao Gu[1,*], Ruxin Ji[1], Wei Shi[1,3], Fangfen Liu[1,3], Hongfu Xie[1,3,4], Ji Li[1,3,4,5], Yicong Liu[1] and Yan Tang[1,3,4]

[1] Department of Dermatology, Xiangya Hospital, Central South University, Changsha, Hunan, China
[2] Department of Dermatology, The Third Affiliated Hospital of Guangzhou Medical University, Guangzhou, China
[3] Hunan Key Laboratory of Aging Biology, Xiangya Hospital, Central South University, Changsha, Hunan, China
[4] National Clinical Research Center for Geriatric Disorders, Xiangya Hospital, Central South University, Changsha, Hunan, China
[5] Key Laboratory of Organ Injury, Aging and Regenerative Medicine of Hunan Province, Central South University, Changsha, Hunan, China
* These authors contributed equally to this work.

Corresponding authors
Yicong Liu, yicongliu96@gmail.com
Yan Tang, ytang_xy@csu.edu.cn

## ABSTRACT

**Objectives:** To investigate the relationship between drinks behavior and female androgenetic alopecia (AGA) and to clarify the mediating effect of sleep behavior on such relationship.

**Methods:** A total of 308 female AGA patients and 305 female normal controls were recruited from the hospital, and questionnaires including drinks behavior and sleep behavior were inquired among them. Blood sugar and blood lipids were detected. Then, the mediating effect of sleep behavior on the relationship between drinks behavior and AGA was analyzed.

**Results:** Female AGA patients presented a higher frequency of intake of sweetened tea drinks. It was found that occasional intake (1–2 times per week; $OR_{adj}$ = 2.130, 95% CI [1.495–3.033]) and frequent intake (3–6 time per week; $OR_{adj}$ = 2.054, 95% CI [1.015–4.157]) of sweetened tea drinks were associated with AGA. The daily sugar intake from soft drinks increased the risk of AGA ($OR_{adj}$ = 1.025, 95% CI [1.003–1.048]), and hyperglycemia was positively associated with alopecia ($OR_{adj}$ = 1.897, 95% CI [1.225–2.936]). In addition, bedtime after 12 pm significantly increased the risk of developing alopecia ($OR_{adj}$ = 2.609, 95% CI [1.020–6.676]). Interestingly, bedtime, instead of daily sugar intake from soft drinks, could mediate the relationship between sweetened tea drinks intake and alopecia.

**Conclusions:** Sweetened tea drinks consumption increases the risk of female AGA, which is mediated by bedtime.

## INTRODUCTION

Female androgenetic alopecia (female AGA) is a common form of nonscarring hair loss that is characterized by progressive thinning and follicular miniaturization (*Bertoli et al., 2020*; *Redler, Messenger & Betz, 2017*). The prevalence of female AGA was reported to be 19% in Caucasians and 5.6% in Asians (*Gan & Sinclair, 2005*; *Paik et al., 2015*). Female AGA manifested mostly with sparseness of the midline hair, which is also termed female pattern hair loss (FPHL). In addition, the fronto-temporal hairline retrusion with/without balding patch on vertex, known as male pattern hair loss (MPHL), is also observed in female AGA patients with a percentage of 0.4–1% (*Ludwig, 2010*; *Su, Chen & Chen, 2013*; *Wang et al., 2010*; *Xu et al., 2009*). The pathogenesis of female AGA is complicated, involving genetic factors, hormonal regulation, oxidative stress, micro-inflammation, and metabolic disorders (*Carmina et al., 2019*; *Erdogan et al., 2017*; *Ramos et al., 2016*; *Redler, Messenger & Betz, 2017*; *Azziz et al., 2009*). Metabolic syndromes (MetS), including abdominal obesity, increased blood sugar (BS), high serum triglycerides (TG), low serum high-density lipoprotein (HDL), and high blood pressure, have been reported to be closely correlated with AGA (*Ahouansou et al., 2007*; *Arias-Santiago et al., 2010*; *El Sayed et al., 2016*; *Kartal et al., 2016*). In addition, central adiposity, MPHL, and polycystic ovary syndrome (PCOS) are defined as classical symptoms of patients with hyperandrogenism, indicating a possibility that androgen might be able to mediate the mechanisms of alopecia and metabolic disorders (*Jeanes & Reeves, 2017*; *Zeng et al., 2020*).

Soft drinks, including carbonated drinks, tea drinks, fruited drinks and dairy drinks, have been considered as one of the dominate sources for daily sugar intake (*Guthrie & Morton, 2000*; *Libuda & Kersting, 2009*). Soft drinks consumption can significantly increase the prevalence or incidence of respiratory diseases (*Al-Zalabani et al., 2019*), urogenital diseases (*Chapman et al., 2019*), mental diseases (*Kashino et al., 2021*), skin diseases (*e.g.*, acne) (*Huang et al., 2019*), and especially metabolic risk factors including obesity, impaired fasting glucose, higher blood pressure and dyslipidemia (*Dhingra et al., 2007*). It was speculated that soft drinks intake might take effect through the glycometabolism or lipid metabolism in the relevant diseases. Based on the close correlation between metabolism-related conditions and AGA, the metabolic pathways underlying AGA have been a research hotspot lately. Current research indicates that the consumption of sugar-sweetened beverages may elevate the risk of androgenetic alopecia in males (*Shi et al., 2023*). Conversely, another studies suggest that such beverages could potentially serve as a protective factor against male-pattern baldness (*Yi et al., 2020b*). However, the relationship between the intake of sugar-sweetened beverages and female pattern hair loss has not yet been elucidated. Hence, we aimed to explore the effect of soft drinks intake on AGA and to clarify the possible mediating mechanism through detection of metabolic indicators.

Other than metabolism-related conditions, soft drinks intake has also been revealed to decrease sleep quality in teenagers (*Boozari, Saneei & Safavi, 2020*; *Sampasa-Kanyinga, Hamilton & Chaput, 2018*). *Vice versa*, health drink behavior can improve sleep quality in women (*Jibril et al., 2022*). Insufficient sleep duration and poor sleep quality are associated

with an increased risk in developing female AGA and are positively correlated with AGA severity (*Liamsombut et al., 2022*; *Yi et al., 2020a*). Thus, sleep behavior might act as a mediating factor on the relationship between soft drinks intake and AGA as well as metabolic factors.

Herein, we recruited 613 female individuals (308 female AGA patients and 305 normal controls) in this case-control study to explore the possible relationship of female AGA with soft drinks behavior, and its possible mediating factors.

## METHODS

### Study subjects and survey

This study was approved by the ethics review board of Xiangya Hospital Central South University (approval number: NO. 201611609) and was conducted from June 2019 to October 2020 in Xiangya Hospital of Central South University, Changsha. Previous studies have shown that the intake rate of soft drinks among Chinese women is 50.8% (*Guo et al., 2018*). Research on the risk factors for hair loss in Korean women has indicated that the odds ratio (OR) for sugar and lipid metabolism ranges from 1.52 to 1.73 (*Kim et al., 2018*). In this experiment, with an exposure rate set at 0.51, an OR of 1.6, a power of 0.8, and a significance level ($\alpha$) of 0.05, the sample size was calculated using the PASS software. The calculation revealed that a group size of 291 individuals per group would be sufficient to meet the experimental requirements (*) (Fig. S1). Accordingly, a total of 613 female participants were recruited, including 308 AGA patients from the department of dermatology and 305 normal controls from the physical examination center. Informed consent statements were obtained from all participants prior to the research work.

AGA was diagnosed and evaluated with the Savins and BASP classifications by two dermatologists independently. The exclusion criteria for AGA patients were: (1) patients with other types of alopecia or polycystic ovary syndrome (PCOS), and (2) patients with a history of or currently undertaking hormone replacement therapy such as testosterone or other hormone related medicine. Normal controls were recruited with matched age according to the AGA group.

The basic participant information, including age, educational background, weight, height, body mass index (BMI), generalized anxiety disorder (GAD), and Patient Health Questionnaire (PHQ), was collected through questionnaires. Comorbidities and drinks behavior were recorded for each participant. The daily sugar intake from soft drinks was calculated as follows: daily sugar intake (grams) = (frequency of carbonated soda consumption per week/7) × sugar content per serving + (frequency of sweetened tea drinks consumption per week/7) × sugar content per serving + (frequency of fruit-flavored drinks consumption per week/7) × sugar content per serving + (frequency of coffee consumption per week/7) × sugar content per serving + (frequency of tea consumption per week/7) × sugar content per serving + (frequency of milk consumption per week/7) × sugar content per serving (*Huang et al., 2019*). The sugar content per serving for different drinks referred to the study by *Huang et al. (2019)*. All laboratory tests were conducted in Xiangya Hospital. Blood sugar (BS) and blood lipids, including triglyceride (TG), total

cholesterol (TC), high-density lipoprotein (HDL) and low-density lipoprotein (LDL), were detected by colorimetry after a 12-h fasting period.

## Statistical analysis

Comparisons between the two groups were performed according to the data type. Continuous data, such as age, BS, TG, TC, HDL and LDL, was presented as mean ± SD, and the between-group differences were tested by t-test. Categorical data, including BMI, education, classification, comorbidities, and drinks behavior, was presented in numbers (%), and the between-group differences were tested by Pearson's $\chi^2$ test.

Logistic regression analysis was used to test the association of influence factors with AGA and AGA severity, including drink behavior (daily sugar intake from soft drinks), and sleep behavior. Diagnosis of AGA and AGA severity were set as the dependent variables. And drink behavior and sleep behavior were set as the independent variables. The crude odds ratio (OR) and the corresponding 95% confidence interval (95% CI) were calculated as model 1. And the $OR_{adj}$ and its corresponding 95% confidence interval were calculated after adjusted by age, height, and weight as model 2.

When the total associations (c) between the soft drinks/daily sugar intake from soft drinks (independent X) and AGA/AGA severity, AGA was significant, the mediation analysis would be considered. The potential mediators (M) were determined to be bedtime, BS and TG. Then the association (a) between X and each of the mediators were estimated, as well as the association between each of the mediators and Y after controlling for X (b) and the direct association (c′) between X and Y after controlling for M. When a and b were both significant, the indirect association (a * b) would be calculated as the mediation effect. The percentage of the mediating effect equals to (a * b)/(a * b + c′). All statistical analyses were performed with SPSS 26.0.

# RESULTS

## Clinical characteristics of female AGA

This study included 308 female AGA patients and 305 female normal controls (Table 1). The mean age of AGA patients and normal controls was 25.91 ± 6.65 and 26.56 ± 4.60, respectively. More than half of the participants had a normal BMI. There were no significant differences in age, BMI and education distribution between the two groups. The prevalence of acne was higher in the female AGA group, while the proportion of uterine tumor was lower in the female AGA group (Table 1). Interestingly, most AGA patients presented an abnormal change especially in the menstrual cycle and menstrual volume (Table 2), though PCOS had been excluded.

The female AGA patients were diagnosed and evaluated with the Savins and BASP classifications. Other than midline changes, a total of 139 (45.2%) patients had hairline involvement, with 44.2% showing mild changes (43.2% M1, 1.0% C1) and 1.0% showing moderate changes (M2) (Table 3). Besides, a majority of patients (233, 75.7%) had mild midline changes (Savins I), while the rest (24.4%) showing moderate-to-severe changes (Savins II, III).

**Table 1 Basic information of participants.**

| | | Ctrl (N = 305) | AGA (N = 308) | p |
|---|---|---|---|---|
| Age, y (mean ± SD) | | 26.56 ± 4.60 | 25.91 ± 6.65 | 0.156 |
| BMI, kg/m$^2$ | | | | |
| | Underweight (<18.5) | 42, 13.8% | 59, 19.2% | 0.120 |
| | Normal (18.5–23.9) | 213, 69.8% | 207, 67.2% | |
| | Overweight (24.0–27.9) | 44, 14.4% | 32, 10.4% | |
| | Obese (≥28.0) | 6, 2.0% | 10, 3.2% | |
| Education | | | | |
| | Elementary education | 2, 0.7% | 1, 0.3% | 0.249 |
| | Medium education | 20, 6.6% | 31, 10.1% | |
| | Advanced education | 283, 92.8% | 276, 89.6% | |
| Comorbidities | | | | |
| | Premature hair graying | 8.50% | 9.40% | 0.778 |
| | Acne | 15.10% | 21.40% | **0.047** |
| | Rosacea | 6.20% | 10.40% | 0.079 |
| | Hirsutism | 0.70% | 0.60% | 0.684 |
| | Hypertension | 0.70% | 0 | 0.247 |
| | Diabetes | 0.30% | 0.30% | 0.748 |
| | Pylori infection | 2.30% | 3.60% | 0.474 |
| | Ovarian tumor | 1.60% | 1.30% | 0.751 |
| | Uterine tumor | 5.90% | 1.30% | **0.002** |
| Depression (PHQ-2) | | | | |
| | PHQ-2 < 3 | 262, 85.90% | 246, 79.90% | **0.048** |
| | PHQ-2 ≥ 3 | 43, 14.10% | 62, 20.10% | |
| Anxiety (GAD-2) | | | | |
| | GAD-2 < 3 | 267, 87.50% | 257, 83.40% | 0.150 |
| | GAD-2 ≥ 3 | 38, 12.50% | 51, 16.60% | |
| Sleep time | | | | |
| | Before 10 pm | 12, 3.90% | 8, 2.60% | **<0.001** |
| | 10–11 pm | 74, 24.30% | 31, 10.10% | |
| | 11–12 pm | 128, 42.00% | 111, 36.00% | |
| | After 12 pm | 91, 29.80% | 158, 51.30% | |
| Sleep duration (hrs) | | 7.04 ± 0.99 | 7.33 ± 0.97 | **<0.001** |

**Note:**
p in bold indicate statistical difference.

Patients with moderate-to-severe alopecia presented a much higher proportion of overweight and obesity than those with mild alopecia did (Table 4). Interestingly, patients with family history showed a higher possibility of hairline involvement (with family history 51.60% *vs.* no family history 41.40%). However, there was no difference in the onset age between the female AGA patients with or without family history (21.98 ± 6.25 years old *vs.* 22.33 ± 6.48 years old, *p* = 0.663).

**Table 2 Menstruation of female AGA and healthy controls.**

|  |  | Ctrl (N = 298) | AGA (N = 305) |  |
|---|---|---|---|---|
| Menstrual cycle |  |  |  |  |
|  | <21 d | 2, 0.7% | 6, 2% | **0.008** |
|  | 21–35 d | 265, 88.9% | 241, 79% |  |
|  | >35 d | 11, 3.7% | 26, 8.5% |  |
|  | Irregular | 20, 6.7% | 32, 10.5% |  |
| Menstrual period |  |  |  |  |
|  | <3 d | 7, 2.3% | 15, 4.9% | **0.048** |
|  | 3–7 d | 268, 89.9% | 279, 91.5% |  |
|  | >7 d | 16, 5.4% | 6, 2% |  |
|  | Irregular | 7, 2.3% | 5, 1.6% |  |
| Menstrual volume |  |  |  |  |
|  | <5 ml | 4, 1.3% | 23, 7.5% | **<0.001** |
|  | 5–80 ml | 259, 86.9% | 260, 85.2% |  |
|  | >80 ml | 21, 7% | 9, 3% |  |
|  | Irregular | 14, 4.7% | 13, 4.3% |  |

**Note:**
$p$ in bold indicate statistical difference.

**Table 3 Two classifications of female AGA (Savins, BASP).**

| AGA classification | N, % |
|---|---|
| Savins (Frontal change, female pattern) |  |
| I-2 | 32, 10.4% |
| I-3 | 117, 38.0% |
| I-4 | 84, 27.3% |
| II-1 | 51, 16.6% |
| II-2 | 19, 6.2% |
| III | 5, 1.6% |
| BASP (Hairline change, male pattern) |  |
| L | 1, 0.3% |
| M0 | 145, 47.1% |
| M1 | 133, 43.2% |
| M2 | 3, 1.0% |
| C0 | 23, 7.5% |
| C1 | 3, 1.0% |

The female AGA group scored higher than the control group in both depression (PHQ-2 score, $1.47 \pm 1.47$ *vs.* $1.26 \pm 1.30$, $p < 0.01$) and anxiety (GAD-2 score, $1.45 \pm 1.41$ *vs.* $1.19 \pm 1.32$, $p < 0.01$) grades. As expected, a significantly higher proportion of female AGA patients manifested an increased intendancy of developing depression (Table 1).

**Table 4 Characteristics of female AGA with different disease severity.**

|  | Mild (Savins I) (N = 233) | Moderate/severe (Savins II and above) (N = 75) | p |
|---|---|---|---|
| Age, y (mean ± SD) | 25.00 ± 5.63 | 28.73 ± 8.56 | 0.001 |
| BMI, kg/m$^2$ |  |  |  |
|     Underweight (<18.5) | 52, 22.3% | 7, 9.3% | 0.001 |
|     Normal (18.5–23.9) | 159, 68.2% | 48, 64.0% |  |
|     Overweight (24.0–27.9) | 17, 7.3% | 15, 20.0% |  |
|     Obese (≥28.0) | 5, 2.1% | 5, 6.7% |  |
| Education |  |  |  |
|     Elementary education | 0 | 1, 1.3% | 0.003 |
|     Medium education | 17, 7.3% | 14, 18.7% |  |
|     Advanced education | 216, 92.7% | 60, 80.0% |  |

## Association between drinks behavior and female AGA

By comparatively analyzing the drinks behavior of female AGA patients and normal controls, it was revealed that female AGA patients tended to have a higher intake frequency of carbonated soda and sweetened tea drinks (Fig. 1, Table S1). However, there were no significant differences in the intake of fruit-flavored drinks, coffee, tea, milk and water between the two groups (Fig. 1, Table S1).

The occasional intake (one to two times a week) of carbonated soda significantly increased the risk of developing AGA by 1.535 times (OR = 1.535, 95% CI [1.071–2.200], $p = 0.019$; $OR_{adj} = 1.525$, 95% CI [1.057–2.202], $p = 0.024$; adjusted by age, height, and weight). Also, frequent intake of sweetened tea drinks might increase the prevalence of AGA. More specifically, occasional intake of sweetened tea drinks increased the risk of developing AGA by 1.978 times (OR = 1.978, 95% CI [1.410–2.777], $p < 0.001$; $OR_{adj} = 2.130$, 95% CI [1.495–3.033], $p < 0.001$; adjusted by age, height, and weight), while frequent intake (3–6 time per week) of sweetened tea drinks increased the risk of developing AGA by 2.044 times (OR = 2.044, 95% CI [1.031–4.055], $p = 0.041$; $OR_{adj} = 2.054$, 95% CI [1.015–4.157], $p = 0.045$; adjusted by age, height, and weight) (Fig. 2).

Moreover, based on the daily sugar intake from soft drinks, it was found that the increase in daily average sugar intake raised the risk of developing AGA by 1.024 times (OR = 1.024, 95% CI [1.003–1.046], $p = 0.024$; $OR_{adj} = 1.025$, 95% CI [1.003–1.048], $p = 0.025$; adjusted by age, height, and weight).

## Changes in sleep behavior and metabolic indicators

The female AGA patients had a tremendous delay in bedtime that more than half of them would not go to bed till midnight, although they might have a longer sleep duration (Table 1). Bedtime after 12 pm significantly increased the risk of developing alopecia, as shown in Fig. 2 (OR = 2.54, 95% CI [0.991–6.506]; $OR_{adj} = 2.609$, 95% CI [1.020–6.676]; adjusted by height and weight).

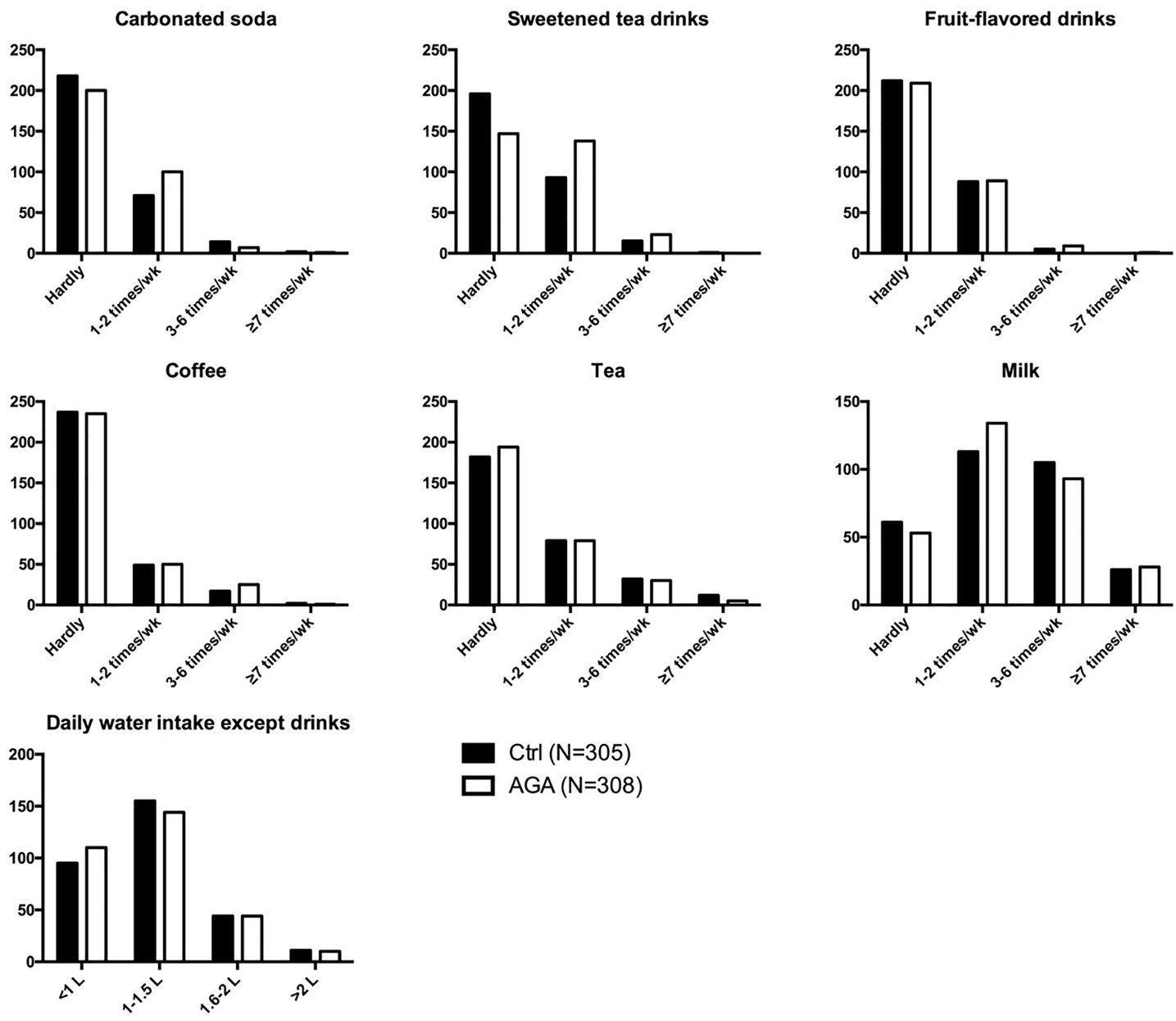

**Figure 1 Drinks behavior in participants.** AGA, androgenetic alopecia.

The BS of female AGA patients was significantly higher than that of controls, whereas the TG level of female AGA patients was lower (Fig. 3, Table S2). Hyperglycemia was found to be positively associated with alopecia (OR = 1.595, 95% CI [1.054–2.412], $p$ = 0.027; $OR_{adj}$ = 1.897, 95% CI [1.225–2.936], $p$ = 0.004; adjusted by age, height, and weight). On the contrary, BS and TG levels of moderate-to-severe AGA patients were significantly higher than those of mild AGA patients (Fig. 3, Table S3), and BS level was positively related to the severity of AGA (OR = 1.005, 95% CI [0.323–1.687], $p$ = 0.004; $OR_{adj}$ = 0.738, 95% CI [−0.025 to 1.502], $p$ = 0.058; adjusted by age, height, and weight).

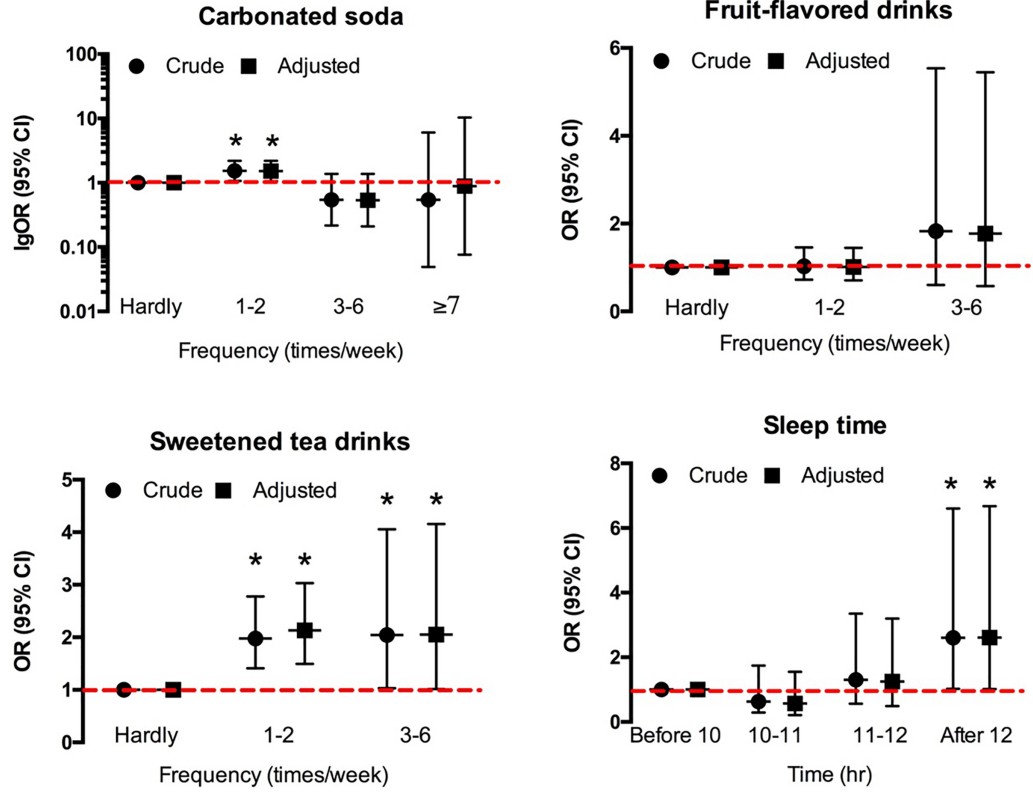

**Figure 2** **The effects of soft drinks intake and sleep on F-AGA.** An asterisk (*) indicates significant statistical difference.

## Mediating factors for the relationship between soft drinks intake and alopecia

Based on the close relationship of soft drinks between metabolism and sleep behavior, the possible mediating effects of metabolic indicators and sleep behavior on the relationship were explored to further clarify the corresponding mechanism. Interestingly, sweetened tea drinks intake was found to regulate AGA by mediating the bedtime, rather than metabolic indicators such as BS and TG (Tables 5, 6, and Fig. 4). The mediating effect of bedtime on the relationship between sweetened tea drinks intake and AGA was 0.64 (95% CI [0.27–1.00]), with a mediating ratio of 17.9%.

## DISCUSSION

According to this study, increased intake of carbonated soda and sweetened tea drinks can raise the risk in developing AGA, accompanied with elevated metabolic indicators and delayed bedtime in alopecia patients.

Acne and AGA were both proposed as metabolic syndromes of the pilosebaceous follicle, which has been found to be closely associated with other metabolic disorders including obesity (*Arias-Santiago et al., 2011*; *Claudel et al., 2018*; *Jeanes & Reeves, 2017*; *Melnik, 2018*). Hyperglycemic food, modern lifestyle nutrition, and soft drinks consumption have been discovered as prominent influencing factors for acne (*Huang*

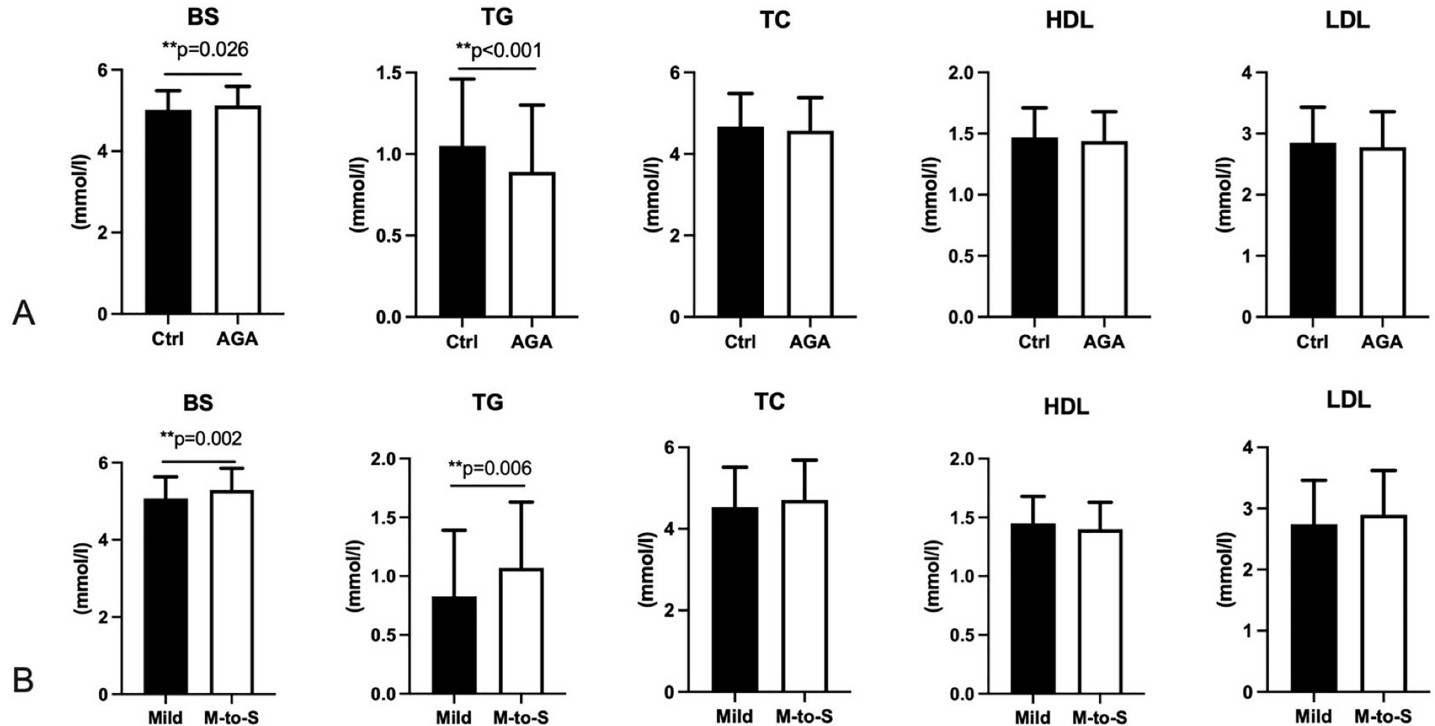

**Figure 3 Blood sugar and blood lipids in participants.** BS, Blood sugar; TG, triglyceride; TC, total cholesterol; HDL, high-density lipoprotein; LDL, low-density lipoprotein. AGA, androgenetic alopecia. Mild, AGA with Savins I; M-to-S, Moderate to severe, AGA with Savins II or above. Two asterisks (**) indicate significant statistical difference.         

**Table 5 Association between intake of drinks behavior, sleep time, AGA and AGA severity.**

| Y | X | M | c′ path β (95% CI)[†] | a path β (95% CI)[†] | b path β (95% CI)[†] |
|---|---|---|---|---|---|
| AGA | Sweetened tea drinks (1–2 per week) | Sleep time | **0.417 [0.248–0.587]** | **0.337 [0.152–0.522]** | **0.637 [0.274–1.000]** |
| | Sweetened tea drinks (3–6 per week) | Sleep time | **0.417 [0.248–0.587]** | **0.922 [0.559–1.286]** | 0.284 [−0.437 to 1.004] |
| AGA severity | Sweetened tea drinks (1–2 per week) | Sleep time | 0.008 [−0.041 to 0.056] | 0.147 [−0.112 to 0.406] | 0.035 [−0.076 to 0.145] |
| | Sweetened tea drinks (3–6 per week) | Sleep time | 0.008 [−0.041 to 0.056] | **0.498 [0.018–0.978]** | −0.117 [−0.324 to 0.089] |

Notes:
[†] Bias-corrected bootstrapped 95% confidence interval.
β in bold indicates statistical significance.
BS, blood sugar; TG, triglyceride.

**Table 6 Association between daily sugar intake from soft drinks, BS, TG, AGA and AGA severity.**

| Y | X | M | c′ path β (95% CI)[†] | a path β (95% CI)[†] | b path β (95% CI)[†] |
|---|---|---|---|---|---|
| AGA | Daily sugar intake from soft drinks | BS | **0.036 [0.010–0.062]** | 0.004 [−0.001 to 0.010] | **0.544 [0.116–0.971]** |
| | | TG | **0.043 [0.016–0.069]** | 0.002 [−0.004 to 0.008] | −0.639 [−1.061 to −0.217] |
| AGA severity | Daily sugar intake from soft drinks | BS | −0.002 [−0.049 to 0.044] | 0.003 [−0.005 to 0.011] | 0.620 [−0.167 to 1.407] |
| | | TG | −0.004 [−0.051 to 0.044] | 0.003 [−0.004 to 0.010] | **0.945 [0.166–1.723]** |

Notes:
BS, blood sugar; TG, triglyceride.
[†] Bias-corrected bootstrapped 95% confidence interval.
β in bold indicates statistical significance.

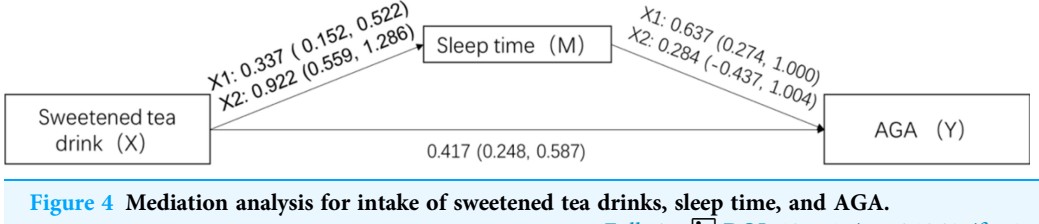

**Figure 4 Mediation analysis for intake of sweetened tea drinks, sleep time, and AGA.**

*et al., 2019*; *Kucharska, Szmurło & Sińska, 2016*). However, evidence on the influence of drinks and food behavior on alopecia is still insufficient. Additionally, AGA is closely correlated with soft drinks consumption based on our results, especially the sweetened tea drinks intake, which contributes to glucose uptake. Androgen receptor sensitivity or androgen excess has been recognized as the dominant etiology for both AGA and acne. Androgen excess was reported to mediate the development of metabolic disturbances in peripheral tissues and visceral organs, thereby contributing to metabolic conditions including obesity and insulin resistance (*Sanchez-Garrido & Tena-Sempere, 2020*). It can be naturally speculated that androgen sensitivity might play an essential role in the metabolic dysfunction among acne or AGA patients, which can lead to a susceptibility to irregular soft drinks behavior.

Sweetened tea drinks, which are generally with high glucose content and sometimes tea polyphenol (TP) and caffeine, present a special influence on alopecia, instead of other soft drinks. The increased glycemic load related to hyperglycemic uptake or sleep disturbance induced by TP or caffeine might be possible intermediary determinants. The daily sugar intake from soft drinks does increase the risk of developing AGA based on our results, but sugar intake may not play a significant role in the influencing mechanism of soft drinks consumption on AGA. This might be explained by a relatively small proportion of sugar intake from the soft drinks compared with the overall food load.

Interestingly, bedtime, instead of sugar uptake, significantly mediates the relationship between soft drinks behavior and female AGA according to our study. TP, caffeine, and extra intake of carbohydrates are all related to sleep behavior, especially delay in bedtime and reduction in sleep duration (*Grandner et al., 2013*). In addition, uncontrolled dietary behaviors are always accompanied with later bedtime and shorter sleep duration in all ages, suggesting a weak healthy intervention on sleep and dietary behavior (*Grummon, Sokol & Lytle, 2020*). As proved, bedtime procrastination significantly raises the risk of developing alopecia. Sleep behavior, which is closely correlated with circadian rhythm regulation, may mediate hair regrowth by affecting the secretion of cortisol and testorsterone (*Hayes, Bickerstaff & Baker, 2010*; *Herzog-Krzywoszanska, Jewula & Krzywoszanski, 2021*). In addition, BAML-1, a central clock gene that regulates circadian rhythm, was also reported to play a part in adjusting hair regrowth (*Watabe et al., 2013*).

However, longer sleep duration, though with a tiny increase, was also discovered to be a risk factor for AGA. A previous study reported that longer sleep duration increased the inclination of frontal and temporal hair loss in female AGA patients (*Gatherwright et al., 2012*). A possible mechanism might be that long sleep duration increases the secretion of

inflammatory factors, such as CRP and IL-6, among which IL-6 can affect hair growth by functioning with dihydrotestosterone (DHT) (*Irwin, Olmstead & Carroll, 2016*; *Kwack et al., 2012*).

Nevertheless, sleep behavior might be an important mediating factor for the association between sweetened drinks intake and AGA. The pathology and etiology behind it need more comprehensive exploration, and an active intervention on both dietary and sleep behavior is of great necessity.

Uterine tumors were found to be negatively correlated with female AGA. As one type of uterine tumors, uterine fibroids are usually accompanied by increased expressions of estrogens, among which 17-α-estradiol can eliminate DHT production to reduce the risk of AGA (*Bulun, 2013*; *Hoffmann et al., 2002*). Further studies are warranted to uncover the association between gynecological cancers and female AGA and to clarify the possible pathophysiology.

This study had several limitations. First, the study sample contained a small proportion of patients with severe alopecia, which might give less information for severe AGA. Meanwhile, the participants recruited were on average in their twenties, indicating that they might be at early onset of alopecia. Hence, the sample should be enlarged in both quantity and age stratification in future. Second, the family history of the participants had not been well recorded regretfully, while might neglect hints for influence of the genetic background. In addition, this was a cross-sectional study that relied on self-reported data. Therefore, a recall bias is unavoidable. It would be hard to anchor the highest weighted factors with amounts of confounding factors in the real-world survey. Basic research might be needed to explore the causal relationship and the mechanism of how sweetened tea drinks and bedtime affect alopecia.

In conclusion, bedtime has a mediating effect on the association between sweetened tea drinks consumption and AGA. The sweetened tea drinks intake should be appropriately restricted and a disciplined sleep behavior is recommended for preventing alopecia.

### Funding
This work was supported by the National Natural Science Foundation of China (grant number 82073467, 91749114). The funders had no role in study design, data collection and analysis, decision to publish, or preparation of the manuscript.

### Grant Disclosures
The following grant information was disclosed by the authors:
National Natural Science Foundation of China: 82073467, 91749114.

### Competing Interests
The authors declare that they have no competing interests.

## Author Contributions

- Shiqi Liu conceived and designed the experiments, performed the experiments, analyzed the data, prepared figures and/or tables, authored or reviewed drafts of the article, and approved the final draft.
- Hao Gu analyzed the data, prepared figures and/or tables, and approved the final draft.
- Ruxin Ji analyzed the data, authored or reviewed drafts of the article, and approved the final draft.
- Wei Shi performed the experiments, authored or reviewed drafts of the article, and approved the final draft.
- Fangfen Liu performed the experiments, authored or reviewed drafts of the article, and approved the final draft.
- Hongfu Xie conceived and designed the experiments, authored or reviewed drafts of the article, and approved the final draft.
- Ji Li conceived and designed the experiments, performed the experiments, authored or reviewed drafts of the article, and approved the final draft.
- Yicong Liu performed the experiments, analyzed the data, prepared figures and/or tables, authored or reviewed drafts of the article, and approved the final draft.
- Yan Tang conceived and designed the experiments, performed the experiments, analyzed the data, prepared figures and/or tables, authored or reviewed drafts of the article, and approved the final draft.

## Human Ethics

The following information was supplied relating to ethical approvals (*i.e.*, approving body and any reference numbers):

The Xiangya Hospital of Central South University granted Ethical approval to carry out the study within its facilities (approval number: NO. 201611609).

## Data Availability

The raw measurements are available in the Supplemental File.

## Supplemental Information

Supplemental information for this article can be found online at http://dx.doi.org/10.7717/peerj.18647#supplemental-information.

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
