# Peer review of "The mediation role of sleep on the relationship between drinks behavior and female androgenetic alopecia"

_PeerJ, doi:10.7717/peerj.18647_

## Round 0.1 · original submission · Major Revisions

Please address all the reviewer comments.

Reviewer 3 ·

Basic reporting

1.Many details mentioned in the text are not reflected in the tables. Additionally, Tables 5 and 6 are missing. Please carefully review and supplement the tables to ensure all relevant data are included.

Experimental design

2.No comment.

Validity of the findings

3.The regression analysis should include more adjustment factors to form different models. In this study, I believe that family history is an important factor that should be included in the analysis.

Additional comments

4.The study is somewhat innovative and fills a gap in the field by focusing on the relationship between sugar-sweetened beverages and female androgenetic alopecia (AGA). However, it is similar to the article "The Association between Sugar-Sweetened Beverages and Male Pattern Hair Loss in Young Men." Improvements are needed in the presentation of figures and tables. The regression analysis should provide more parameters, such as p-values and model fit indices. For the mediation analysis, please provide specific values and confidence intervals for each path coefficient in the results section.

·

Basic reporting

Line 1: The mediation role of sleep on the relationship between drinks behavior and female androgenetic alopecia is different than title of sent manuscript to be reviewed (The association between soft drinks behavior and female androgenetic alopecia).
You must mention how to calculate the sample size.
You should include a flow chart.

Experimental design

no comment

Validity of the findings

no comment

Additional comments

no comment

Reviewer 6 ·

Basic reporting

The writing and the reporting of the manuscript is clear, the structure is well established. Authors have explained the purpose of the study in the introduction and it is easy to follow. Literature references are okay.

Experimental design

The study design is okay, the clinical information collected are reasonable. The statistical tests are some what questionable from my opinion.

The statistical method (logistic regression) was not clearly explained in the manuscript. The authors used many symbols to represent variables, which caused confusion in reading the paragraph. The process of variable selection should be described in plain English. "Mediators" are adjusted variables, and authors cannot assume readers have the knowledge of the formula for logistic regression, thus using symbols like X(b) is not a good idea. The paragraph should be rephrased. A better flow would be to explain that first there will be univariate logistic regression performed on a, b, c variables, and then if significant, will add more variables to adjust the model. This leads to another question, why would the "mediators" be added after significance? If the variable is already adjusted, why adjusted by another set of variables after it tested significance? Does this create unfairness/multiple testing to the model?

Validity of the findings

The findings of the study are very interesting.
Something I found not aligned:
1. Section 3.2: "The occasional intake of carbonated soda significantly increased the risk of developing AGA by 1.535 times", but the adjusted OR is 1.525 in the text. Please double check.
2. Section 3.4, the authors explained mediating effects of metabolic indicators and sleep behavior might have a corresponding mechanism. Have the authors think about interaction effect in the model? While adding more adjusted variables that correspond with others, there might be interactions.
3. The limitations of the study are summarized in a very short paragraph. It would be better the authors can elaborate and point future directions for people in the similar research field, or in general about this topic.

·

Basic reporting

The author discussed potential mechanisms in the discussion section, including how sugar intake from soft drinks and sleep behavior influence the development of F-AGA.
These discussions are valuable, yet the author could explore other potential mechanisms more thoroughly, such as the impact of caffeine and other components in soft drinks.

Experimental design

The author did not clearly articulate the existing research on the association between soft drink consumption and F-AGA, nor how his/her study addresses this knowledge gap. It is recommended to include this information in the introduction.

Validity of the findings

The author should detail the criteria for selecting AGA patients and normal controls, and explain how to ensure the comparability between the two groups.
The description of the author's statistical analysis methods is insufficiently clear. For instance, the author employed logistic regression analysis to assess the association between soft drink consumption and F-AGA; however, the method for handling interactions among multiple variables and model selection was not adequately explained. Additionally, the article refers to mediation analysis without detailing its execution.
I suggest that the author further elaborate on each step of the statistical analysis.

Additional comments

There are no additional aspects to be included.

---

## Round 0.2 · Minor Revisions

Please address the final comments from the reviewer, asking that you expand upon the limitations and potential for future extensions of your work

Reviewer 6 ·

Basic reporting

The writing and the reporting of the manuscript is clear, the structure is well established. Authors have explained the purpose of the study in the introduction and it is easy to follow. Literature references are okay.

Experimental design

The statistical method (logistic regression) section is updated. It is clearer now what is the dependent variable and what are the independent variables. Mediators are also explained in section 2.2.

Validity of the findings

Most previous questions are answered by authors. However, the limitation of the study are still summarized in a very short paragraph. It would be better if the authors can elaborate on that and proactively thinking about how studies like this can be further investigated and minimize limiations.

·

Basic reporting

no comment

Experimental design

no comment

Validity of the findings

no comment

Additional comments

In response to the issues raised in the previous manuscript, the author has made adjustments to the content, including improving the article format, providing detailed explanations of experimental design and data, etc.
Therefore, I believe that the revised manuscript can meet the requirements for publication.

---

## Round 0.3 · accepted · Accept

The authors have adequately addressed all of the review comments, and the manuscript is now appropriate for publication.

My only suggestion is to undertake one final revision of the manuscript to ensure any remaining errors associated with the use of English are corrected.